# Development of a Gold Nanoparticle-Linked Immunosorbent Assay of Staphylococcal Enterotoxin B Detection with Extremely High Sensitivity by Determination of Gold Atom Content Using Graphite Furnace Atomic Absorption Spectrometry

**DOI:** 10.3390/pharmaceutics15051493

**Published:** 2023-05-13

**Authors:** Chaojun Song, Yutao Liu, Jinwei Hu, Yupu Zhu, Zhengjun Ma, Jiayue Xi, Minxuan Cui, Leiqi Ren, Li Fan

**Affiliations:** 1School of Life Science, Northwestern Polytechnical University, Xi’an 710072, China; cj6005@nwpu.edu.cn; 2Department of Pharmaceutical Analysis, School of Pharmacy, Air Force Medical University, Xi’an 710032, China; liuyutao@fmmu.edu.cn (Y.L.); hujinwei@fmmu.edu.cn (J.H.); zhuyupupapa@fmmu.edu.cn (Y.Z.); mazhengjun@fmmu.edu.cn (Z.M.); 0518xjy@fmmu.edu.cn (J.X.); cuiminxuan2521@fmmu.edu.cn (M.C.)

**Keywords:** gold nanoparticles, gold nanoparticle-linked immunosorbent assay, staphylococcal enterotoxin B

## Abstract

Highly sensitive staphylococcal enterotoxin B (SEB) assay is of great importance for the prevention of toxic diseases caused by SEB. In this study, we present a gold nanoparticle (AuNP)-linked immunosorbent assay (ALISA) for detecting SEB in a sandwich format using a pair of SEB specific monoclonal antibodies (mAbs) performed in microplates. First, the detection mAb was labeled with AuNPs of different particle sizes (15, 40 and 60 nm). Then the sandwich immunosorbent assay for SEB detection was performed routinely in a microplate except for using AuNPs-labeled detection mAb. Next, the AuNPs adsorbed on the microplate were dissolved with aqua regia and the content of gold atoms was determined by graphite furnace atomic absorption spectrometry (GFAAS). Finally, a standard curve was drawn of the gold atomic content against the corresponding SEB concentration. The detection time of ALISA was about 2.5 h. AuNPs at 60 nm showed the highest sensitivity with an actual measured limit of detection (LOD) of 0.125 pg/mL and a dynamic range of 0.125–32 pg/mL. AuNPs at 40 nm had an actual measured LOD of 0.5 pg/mL and a dynamic range of 0.5 to 128 pg/mL. AuNPs at 15 nm had an actual measured LOD of 5 pg/mL, with a dynamic range of 5–1280 pg/mL. With detection mAb labeled with AuNPs at 60 nm, ALISA’s intra- and interassay coefficient variations (CV) at three concentrations (2, 8, and 20 pg/mL) were all lower than 12% and the average recovery level was ranged from 92.7% to 95.0%, indicating a high precision and accuracy of the ALISA method. Moreover, the ALISA method could be successfully applied to the detection of various food, environmental, and biological samples. Therefore, the successful establishment of the ALISA method for SEB detection might provide a powerful tool for food hygiene supervision, environmental management, and anti-terrorism procedures and this method might achieve detection and high-throughput analysis automatically in the near future, even though GFAAS testing remains costly at present.

## 1. Introduction

Staphylococcal enterotoxin B (SEB) is one of the major staphylococcal enterotoxins (SEs) which often cause food derived poisoning even exposed to high temperature or proteolytic enzymes. In humans, the half-lethal dose (LD50) and half-effective dose (ED50) of SEB were only 20 ng/kg and 0.4 ng/kg [1,2]. According to the SEB toxicity, stability, ease of preparation, and ease of spreading by water, food and aerosol [3,4,5], it was listed as a category B potential biological weapon by the Center for Disease Control and Prevention (CDC) [6,7]. Thus, development of SEB detection methods with high sensitivity and specificity are very important in food safety or in the war against biological weapons which are often used in terrorism.

Several conventional methods have been established for detecting SEB, including chemiluminescent immunoassay [8], electrochemiluminescence immunoassay [9], fluorescent immunoassay [10], enzyme-linked immunosorbent assay [11], and lateral flow assay [12]. Although these methods have played important roles in the past decades for quantitative analysis of SEB, their sensitivity still cannot meet the requirements of warning of SEB contamination at the early stage. In recent years, great efforts have been made to improve the sensitivity of SEB detection, including various biosensors [13,14,15,16]. However, especially the high equipment prices have made these detection methods difficult to scale up. The necessity to develop highly sensitive and relatively simple SEB detection methods remains of great importance.

GFAAS is a widely used technique for measuring metal ions with high sensitivity, low sample consumption and high detection speed, and has been applied in the detection field in the early stages combined with immunoassay [17,18]. Based on this, we present a novel method based on sandwich immunoassay of SEB by a highly specific monoclonal antibody (mAb) pair of SEB and the detection mAb was labeled with AuNPs as indicator agents, named as ALISA. This could increase the actual measured LOD of SEB to 0.125 pg/mL, and has been considered more sensitive than ELISA and even radio immunoassay (RIA) [19]. The capture antibody was coated on the microplate, and SEB standard or various matrices containing a known concentration of SEB was added. After complete reaction and washing, the detection antibody, which combined with the AuNPs, was added. After washing, the AuNPs were digested with aqua regia, and the content of gold atoms was determined by graphite furnace atomic absorption spectrometry. By drawing the standard curve between the gold atom content and the corresponding SEB concentration, the SEB concentration in the samples could be conveniently obtained.

In immunoassays based on enzyme-catalyzed substrates (tetramethylbenzidine, luminol, etc.), the enzyme’s catalysis on the substrate is greatly affected by temperature and reaction time [20,21], especially when the time of substrate addition is inconsistent, which will affect the accuracy of the results to different degrees. The content of gold atoms in AuNPs after aqua regia digestion is a constant value, which is hardly affected by temperature and digestion time. Therefore, the immunoassay using AuNPs as an indicator is more accurate in determining the results. Moreover, in view of the high stability of gold atoms, the digested samples could be stored for a long time and repeatedly tested at any time, which is completely impossible to achieve with other immunological assays. Furthermore, with the improvement of the instrument of atomic absorption spectrometry, the analysis method presented in this study could automatically achieve detection and high-throughput analysis.

## 2. Materials and Methods

### 2.1. Materials and Instruments

Gold (III) chloride hydrate (Au ≥ 48%), sodium citrate tribasic dihydrate (Na_3_Ct∙2H_2_O), hydroxylamine hydrochloride (NH_2_OH∙HCl), N-hydroxysuccinimide (C_4_H_5_NO_3_), N-(3-dimethylaminopropyl)-N′-ethylcarbodiimide hydrochloride (C_8_H_17_N_3_∙HCl) and sodium hydroxide (NaOH) were purchased from Sigma (St. Louis, MO, USA). HS-PEG_1000_-COOH was purchased from Xi’an Ruixi Biological Technology Co., Ltd. (Xi’an, China). Casein sodium salt was purchased from Shanghai Aladdin Biochemical Technology Co., Ltd. (Shanghai, China). Citric acid, NaCl, KCl, Na_2_HPO_4_∙12H_2_O, KH_2_PO_4_, H_2_O_2_ (30%), HNO_3_ (68%), HCl (37%), and Tween-20 were purchased from Damao Chemical Reagent Factory (Tianjin, China). Milli-Q deionized water was used throughout the experiments. For the cleaning of the glasswork, we employed a solution of HNO_3_ and HCl (3:1) in fresh water and then used it within 1 h. SEB protein and the mAb pair for SEB detection (FMU-SEB-2 as capture mAb and FMU-SEB-1 as detection mAb) were gifts of Professor Boquan Jin from the Airforce Millitary Medical University [22]. All the experiments of SEB detection were carried out in a biological safety cabinet and the waste liquid was collected and treated in an incinerator.

Transmission electron microscopy (TEM) images were obtained using a TECNAI G2 Spirit Biotwin (FEI, Wikiwand, OR, USA). UV–vis adsorption spectra of the samples were recorded with a Synergy LX Multimode Reader (BioTek, Winooski, VT, USA). Fourier transform infrared spectra (FTIR) were obtained by using a Nicolet IS50 (Thermo Fisher Scientific, Waltham, MA, USA). The gold atoms were detected using a ZEEnit 700P (Analytik Jena, Jena, Germany).

### 2.2. Preparation of Gold Nanoparticle (AuNP) Seeds

Average particle sizes of 15 nm and 28 nm AuNPs were prepared by using the citrate reduction protocol previously reported [23,24]. First, a stock solution of 41.89 mM HAuCl_4_∙xH_2_O was prepared in Millipore water. Briefly, 597 μL of 41.89 mM stock solution was added to 120 mL Millipore water under vigorous stirring. After the solution was boiled in a silicone oil bath (135 °C), 2 mL of 62.5 mM sodium citrate solution was quickly added to the mixture and stirred until red wine colored AuNP colloids formed. Then, the solution was maintained for 10 min at boiling temperature and removed from the heating mantle. Stirring was continued for another 15 min. The AuNP colloids of 15 nm diameter were characterized by transmission electron microscopy (TEM). Similarly, 730 μL of 41.89 mM stock solution was added to 120 mL Millipore water under vigorous stirring. Then, the mixed solution was added to 12 mL of 40 mM sodium citrate solution. The AuNP colloids of 28 nm diameter were characterized by TEM.

### 2.3. Preparation of the Larger Particle Size Colloidal Gold

The prepared gold seed particles were used to synthesize gold particles larger than 20 nm following the procedure developed by Xu et al. [25]. The larger AuNPs with a diameter of 40 nm and 60 nm were prepared by the following procedure: 10 mL of 15 nm or 28 nm gold nanoparticle seeds was added to 90 mL of deionized water (“Millipore”), and the mixture was stirred for 1 min. The mixed solution was first added with 1 mL of 0.1 M hydroxylamine hydrochloride solution, and after stirring for one minute, 0.7 mL of 25.4 mM chloroauric acid solution was slowly added dropwise. Interestingly, it was found that the color of the solution gradually changed from light pink to wine red. The resulting gold colloids were stored at 4 °C in the dark to minimize photoinduced oxidation. The AuNP colloids of 40 nm and 60 nm diameter were characterized by TEM.

### 2.4. Preparation of AuNPs/HS-PEG-COOH/FMU-SEB-1 Detection mAb Bioconjugate

Surface modification of AuNPs was performed by the direct mixing of an HS-PEG-COOH solution with the AuNP colloid, as described by Mohd-Zahid et al. [26], with minor modifications. First, 20 mL of colloidal gold solution was adjusted to pH 10 with NaOH solution. 2 mL of 20 mg/mL HS-PEG_1000_-COOH solution was added to AuNP colloid, then suspended until homogeneous and incubated at 4 °C for 24 h to obtain AuNP/HS-PEG-COOH solution. The excess HS-PEG_1000_-COOH was removed by placing 22 mL of solution in a dialysis tubing in 5 L of deionized water for 24 h and replacing the deionized water twice. The modified AuNPs were kept at 4 °C.

The bioconjugate was prepared according to the previous procedure [27]. SEB detection mAb (FMU-SEB-1) was used to label the modified AuNPs. Reaction raw mass ratio was as follows: AuNP/HS-PEG-COOH:FMU-SEB-1 (5:1). Briefly, a mixture of 40 μL of EDC (50 mM) and 40 μL of NHS (50 mM) were added to 500 μg PEG-modified AuNPs and stirred for 2 h at room temperature, followed by centrifuging the mixture to remove the unreacted NHS, EDC, and byproducts. The solution was redispersed in 1 mL of 5 mM phosphate buffer (PB, pH 7.4). Then, 100 μg FMU-SEB-1 was added and the reaction was maintained overnight at 4 °C. After washing three times with deionized water, the AuNP/HS-PEG-COOH/FMU-SEB-1 was centrifuged to remove the non-binding mAb and resuspended with phosphate-buffered saline (PBS, 8 mM phosphate buffer with 145 mM NaCl, pH 7.4) containing 0.02% NaN_3_ (*w*/*v*) and stored at 4 °C for further use.

### 2.5. SEB Detection Using SEB Specific mAb Pair by ALISA

The procedures of ALISA are shown in Figure 1. In brief, SEB capture antibody (FMU-SEB-2) was diluted to 10 μg/mL with 0.05 M carbonate buffer (pH 9.6) and 100 μL diluted FMU-SEB-2 was added to a 96-well microplate, followed by incubating overnight at 4 °C. The microplate was washed three times with PBS containing 0.05% Tween 20 (*v*/*v*) (PBST, pH 7.4), and then 100 µL/well of serially diluted SEB with dilution buffer (0.5% casein sodium salt in PBS, pH 7.4) or various matrices containing SEB was added. The microplate was incubated at 37 °C for 1 h and then washed three times with PBST to remove the unbounded SEB. Afterwards, AuNPs-labeled FMU-SEB-1 dilution solution (100 µL/well) was added to the wells and incubated at 37 °C for 1 h. After three washes with PBST, the AuNPs were dissolved by adding aqua regia (100 μL/well) to the microporous plate for 15 min. The solution in the well was collected with a micropipette and transferred to a polypropylene tube. In order to avoid corrosion of the graphite furnace, the solution was boiled in a water bath for 30 min to remove volatile hydrochloric acid, the volume was adjusted to 100 μL with deionized water, and the gold atoms were detected on the graphite furnace.

### 2.6. Sample Preparation

We used different approaches to prepare different matrices. In brief, for the non-soluble solid and semi-solid matrices such as roast beef, peanut butter, soybean paste, cured ham, ketchup, blueberry jam, etc., weigh 10 g of each in a mortar, cut with scissors, add a small amount of dilution buffer (0.5% sodium casein in PBS, pH 7.4), then grind thoroughly in the mortar to a paste and collect the contents in a centrifuge tube. The mortar is rinsed 3 times with an appropriate amount of diluent (0.5% sodium casein in PBS, pH 7.4) and collected in the previous centrifuge tube. Finally, the final pH value is adjusted to 7.4 and the volume adjusted to 20 mL. Place on a vertical mixer for 1 h to allow full leaching of contents. Centrifuge at 1000× *g* for 10 min at 4 °C to remove particulate matter. The supernatant is transferred to a new centrifuge tube, and the supernatant used to dilute the SEB to final concentrations of 0.125, 1, 4, 8, 16, and 32 pg/mL. For the liquid matrix such as orange juice, apple juice, river water, human serum, milk, etc., take 10 mL of each, dilute with the appropriate diluent (0.5% sodium casein in PBS, pH 7.4) to the final concentration of 50% and adjust the final pH value to 7.4, and place on the vertical suspension instrument for 1 h to fully leach the contents. Centrifuge at 1000× *g* for 10 min at 4 °C to remove particulate matter. The supernatant is transferred to a new centrifuge tube, and the supernatant used to dilute SEB to final concentrations of 0.125, 1, 4, 8, 16, and 32 pg/mL.

## 3. Result and Discussion

### 3.1. Characterization of Gold Nanoparticles of Different Sizes

Four different sized gold nanoparticles were prepared. TEM data (Figure 2) showed that the particles were spatially separated and the average diameter of colloidal gold particles was almost the same for one sample: approximately 15 nm, 40 nm, 60 nm, and 80 nm in diameter, respectively. Figure 2d demonstrated that the 80 nm colloidal gold no longer had a spherical shape but grew toward an ellipse shape and had a certain tendency to aggregate. Therefore, it could not be used for the following studies. Particle size analysis software was used to measure the average particle size of 100 nanoparticles in each batch of samples. The particle size distribution diagram (Figure 3) displayed that the particle size distribution of each batch of samples was uniform, showing a concentrated distribution trend. The visible spectroscopy data in Figure 4 showed that the optimum absorption of gold nanoparticles at 15 nm, 40 nm, and 60 nm are at 519.5 nm, 523.5 nm, and 533 nm, respectively, indicating that the larger sized gold nanoparticles had a small red shift compared to the smaller ones. This is because the wavelength of the surface plasma absorption increases with the increase of the particle size of the nanoparticles [28,29].

### 3.2. Characterization of AuNP/HS-PEG-COOH and AuNP/HS-PEG-COOH/FMU-SEB-1

The SH-PEG-COOH can be a link to connect AuNPs and FMU-SEB-1. The formation of AuNP/SH-PEG-COOH is based on the interaction between the terminal -SH group of PEG and Au to undergo the alkanethiol reaction, which is directly emulsified on the surface of the AuNP. In fact, grafting of sulfated PEGs has been widely reported in the literature, giving the particles high colloidal stability by forming strong Au-S bonds due to hydrophilic and sterically hindered binding [30,31,32]. In addition, the -COOH group of PEG formed a peptide bond with the -NH_2_ group of FMU-SEB-1, which together formed the AuNP/SH-PEG-COOH/FMU-SEB-1 bioconjugate.

The visible absorption spectra confirmed the formation of AuNP/SH-PEG-COOH and AuNP/SH-PEG-COOH/FMU-SEB-1 bioconjugate, as shown in Figure 5. Black lines in Figure 5a–c showed the visible absorption spectra of AuNPs at 15 nm, 40 nm, and 60 nm. Figure 5 (red line) displayed the visible absorption spectra of HS-PEG-COOH modified AuNPs, which showed that the visible absorption spectra of the PEG-modified AuNPs were all red-shifted. The resonance wavelength and bandwidth of AuNPs are dependent on the particle size and shape, the refractive index of the surrounding medium, and the temperature. This shift shows the changes of the dielectric nature surrounding the AuNPs due to the association of HS-PEG-COOH molecules with gold nanoparticles to form a stable covalent bond Au-S [33]. The blue lines in Figure 5 showed the visible absorption spectra of antibody modified AuNPs. Similarly, a slight red shift in the extinction maximum was observed upon adsorption of FMU-SEB-1 onto the AuNPs. This spectral shift resulted from a change in the local refractive index due to the protein adlayer and is consistent with previous reports [34]. The displacement lengths of nanoparticles of different sizes are shown in Table 1.

The TEM data (Figure 6a–c) showed that there was a transparent aperture, since the gold nanoparticles that cross-linked with protein showed a shell-core structure after TEM magnification, and the protein formed a layer of hydration. The less dense protein presented a transparent aperture under the penetration of the electron beam. Phospho-tungstic acid combined with the positively charged groups in the proteins to form a high electron-density substance, which appeared as a dark black color under electron beam penetration (Figure 6d–f). In addition, FTIR characterization of gold nanoparticles before and after modification was performed, as shown in Appendix A. These results indicated that the SEB detection mAb had been successfully labeled to the gold nanoparticle surface.

### 3.3. Establishment of ALISA Method and Sensitivity of SEB Detection

AuNPs with different particle sizes have different diffusivity, stability in solvent, specific surface area, and gold atom content of a single AuNP. Therefore, the particle size of AuNPs might have an influence on the sensitivity of SEB detection in ALISA. Theoretically, within a certain range, the larger the particle size of AuNPs, the higher is the gold atom content of a single AuNP, and the more obvious is the amplification effect of ALISA detection. In this study, four AuNPs with different particle sizes were synthesized, which were 15 nm, 40 nm, 60 nm, and 80 nm, respectively. However, AuNPs with 80 nm had poor dispersion in water, and TEM results showed that most AuNPs were aggregated, so they could not be used in ALISA detection (Figure 2d).

First, the optimal dilution ratio of AuNPs-labeled FMU-SEB-1 was determined. The peak absorbance of AuNPs-labeled FMU-SEB-1 was adjusted to 2.0, and then diluted with diluents (0.5% sodium casein in PBS, pH 7.4) of 1:20, 1:40, 1:80, 1:160, and 1:320, respectively. The SEB concentration was adjusted to 0, 0.01, 0.1, 1, 5, and 20 pg/mL to determine the optimal concentration of AuNPs-labeled FMU-SEB-1 with the best signal-to-noise (S/N) ratio. As shown in Figure 7a, AuNPs-labeled FMU-SEB-1 with a particle size of 15 nm had the highest S/N ratio (2.33) when the SEB concentration was 5 pg/mL and the dilution ratio was 1:160 (positive when S/N ratio > 2.1). AuNPs-labeled FMU-SEB-1 with a particle size of 40 nm had the highest S/N ratio of 4.94–151.56 when the SEB concentration was 1–20 pg/mL and the dilution ratio was 1:80 (Figure 7b). AuNPs-labeled FMU-SEB-1 with a particle size of 60 nm had the highest S/N ratio of 2.01–616.38 at SEB concentration of 0.1–20 pg/mL and dilution of 1:80 (Figure 7c), indicating that AuNPs-labeled FMU-SEB-1 with a particle size of 60 nm has the highest detection sensitivity of about 0.1 pg/mL. Then, serial concentrations of SEB (0.125–1280 pg/mL) were detected by AuNPs-FMU-SEB-1 with the optimal dilution, and the standard curve was drawn between the gold atomic content and the corresponding SEB concentration. The typical standard curves of SEB detection using the ALISA method are shown in Figure 8.

AuNP conjugates with different particle sizes had different detection sensitivities in the ALISA method. The 60 nm AuNPs showed the highest sensitivity, with an actual measured LOD of 0.125 pg/mL, and a dynamic range of 0.125–32 pg/mL. The 40 nm AuNPs showed the next highest sensitivity, with an actual measured LOD of 0.5 pg/mL, and a dynamic range of 0.5–128 pg/mL. AuNPs of 15 nm had the lowest sensitivity, only 5 pg/mL, and a dynamic range of 5–1280 pg/mL (Figure 8). Table 2 lists the newly established methods for SEB detection, including their linear ranges, detection limits, detection times, etc. Noticeably, compared to other methods, ALISA is highly sensitive. Although the current detection steps in the laboratory stage are relatively complicated, with the improvement of analytical instruments, the ALISA method might achieve automatic and high-throughput analysis, which is of great importance in the applications for the actual detection of SEB.

### 3.4. Precision and Accuracy of SEB Detection Using ALISA

Traditionally, the precision of detection methods was determined by intra-assay and inter-assay. First, SEB was diluted to three concentrations with dilution buffer (0.5% sodium casein in PBS, pH 7.4), which was 20 pg/mL (high concentration), 8 pg/mL (medium concentration), and 2 pg/mL (low concentration). For intra-assay, eight samples of each concentration were analyzed in triplicate by ALISA using AuNP conjugates of 60 nm in the same batch and the CV of SEB detection was calculated. For inter-assay, eight samples of each concentration were performed on different days in triplicate by ALISA and the CV of SEB detection was calculated. The CV for the intra-assay and inter-assay precision determinations ranged from 4.4% to 7.2% and from 10.8% to 12.3%, respectively (Table 3), indicating high reproducibility of the ALISA method. The original data has been placed in Appendix A.

The accuracy of the ALISA detection method was verified by recovery experiment. Briefly, three concentrations of SEB were added into the diluent (0.5% casein sodium salt in PBS, pH 7.4) as the detection samples (20 pg/mL, 8 pg/mL, and 2 pg/mL, respectively), and then the SEB concentrations in the samples were determined by the ALISA method using AuNP conjugates of 60 nm to evaluate the accuracy. The recovery experiment results of ALISA detection of SEB showed that the average recovery level ranged from 92.7% to 95.0% (Table 4), which indicated that the ALISA method for the detection of SEB had high accuracy and might be promising for applications of detection of SEB in food, environmental, or biological samples. The original data has been placed in Appendix A.

### 3.5. Recoveries of SEB Detection Using ALISA in Various Matrices

Although the sensitivity, repeatability, and accuracy of SEB detection by the ALISA method were satisfactory, its applicability and accuracy in practical applications still needed to be evaluated. In this study, we tested the recoveries of SEB by the ALISA method in a variety of food, environmental and biological substrates using a recovery test to assess its applicability and accuracy. The final concentrations of spike SEB were 0.125, 1, 4, 8, 16, and 32 pg/mL, respectively. SEB was detected by the ALISA method using AuNPs conjugates of 60 nm in food matrices (cured ham, roast beef, peanut butter, ketchup, blueberry jam, soybean paste, orange juice, and milk), an environmental sample (river water), and a biological sample (human serum). The recovery test results of various food matrices, the environmental sample, and the biological sample are shown in Figure 9. Even at the very low concentration of 0.125 g/mL, the recovery rate of SEB was still very high, ranging from 81.30% to 104.83%. These results indicated that the ALISA method was suitable for the detection of SEB in food substrates, environmental samples, and biological samples. Moreover, its extreme high sensitivity is very important for early warning of SEB biological threats.

## 4. Conclusions

In this study, we presented an ALISA method for the detection of SEB, with an extremely high detection limit to 0.125 pg/mL and a dynamic range of 0.125–32 pg/mL. Detection of SEB by ALISA could be applied to detection in a variety of food substrates, environmental samples, and biological samples. Leaving aside the high testing costs of GFAAS, extremely high SEB detection sensitivity was of great significance for the early warning of SEB biological threats, especially for food hygiene supervision, environmental management, anti-terrorism procedures, and other fields.

## Figures and Tables

**Figure 1 pharmaceutics-15-01493-f001:**
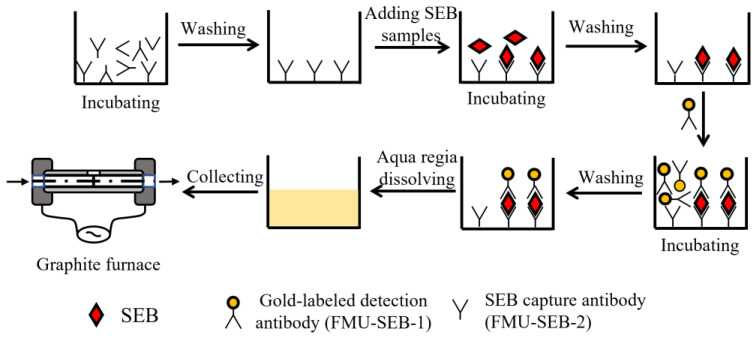
Flow chart of the gold nanoparticle-linked immunosorbent assay.

**Figure 2 pharmaceutics-15-01493-f002:**
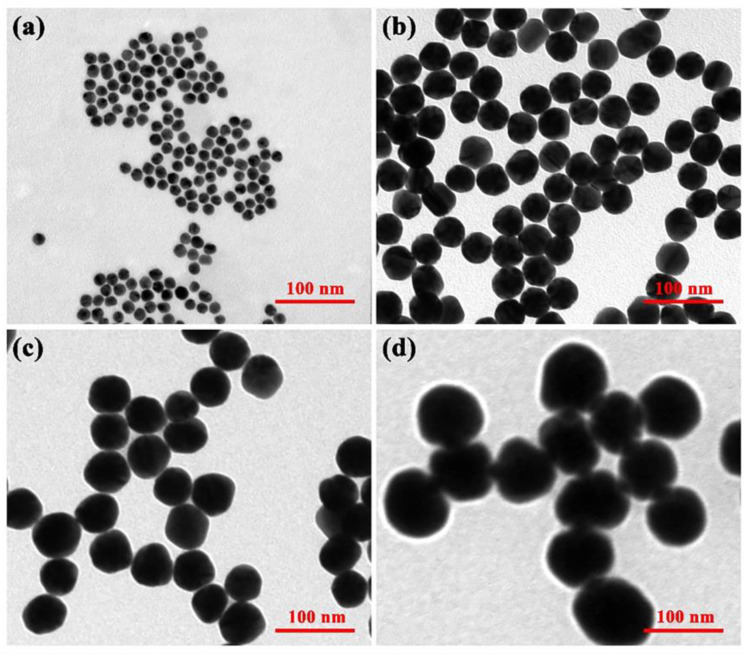
TEM image of (**a**) 15 nm AuNPs, (**b**) 40 nm AuNPs, (**c**) 60 nm AuNPs, and (**d**) 80 nm AuNPs.

**Figure 3 pharmaceutics-15-01493-f003:**
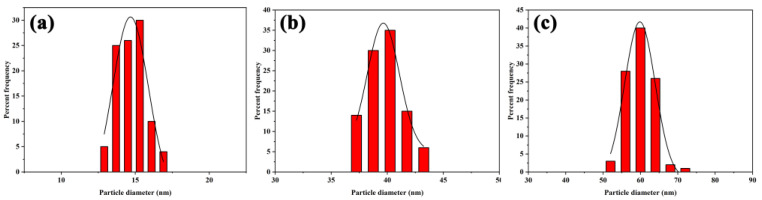
Size distribution of 100 GNPs with average diameter of (**a**) 14.8 ± 0.9 nm, (**b**) 39.6 ± 1.6 nm, and (**c**) 60.1 ± 3.7 nm.

**Figure 4 pharmaceutics-15-01493-f004:**
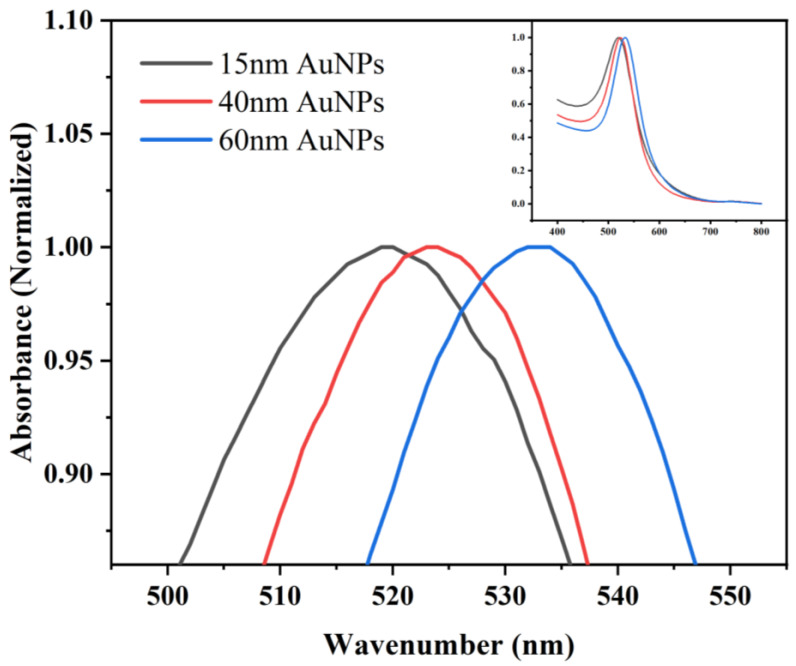
The characteristics of AuNPs of different sizes with visible spectroscopy.

**Figure 5 pharmaceutics-15-01493-f005:**
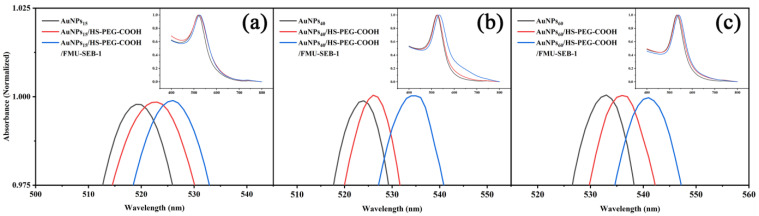
Characterization of unconjugated and functionalized AuNPs by visible spectrophotometry to monitor surface modification. Extinction spectra are provided for AuNPs, HS-PEG-COOH-modified AuNPs, and HS-PEG-COOH-modified AuNPs combined with FMU-SEB-1 mAb of 15 nm (**a**), 40 nm (**b**) and 60 nm (**c**). The inset shows a visible spectrum in the wavelength range 400–800 nm.

**Figure 6 pharmaceutics-15-01493-f006:**
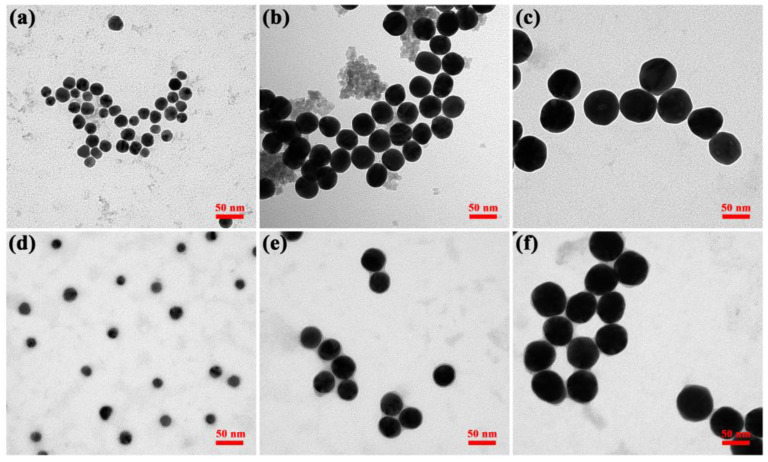
TEM images of (**a**) 15 nm, (**b**) 40 nm, and (**c**) 60 nm antibody modified AuNPs; (**d**–**f**) are the images of (**a**–**c**) after phosphotungstic acid staining, respectively. The scale bar is 50 nm.

**Figure 7 pharmaceutics-15-01493-f007:**
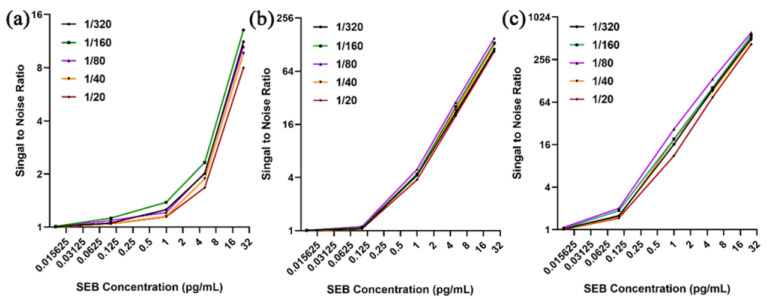
Optimization of dilution ratio of AuNP-labeled FMU-SEB-1. (**a**) The optimal dilution ratio of 15 nm AuNPs-labeled FMU-SEB-1 was 1:160. (**b**) The optimal dilution ratio of 40 nm AuNPs-labeled FMU-SEB-1 was 1:80. (**c**) The optimal dilution ratio of 60 nm AuNPs-labeled FMU-SEB-1 was 1:80.

**Figure 8 pharmaceutics-15-01493-f008:**
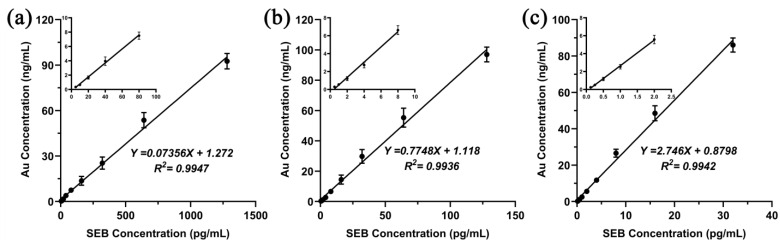
Typical standard curve of ALISA with different AuNP size. (**a**) AuNPs of 15 nm had an actual measured LOD of 5 pg/mL and a dynamic range of 5–1280 pg/mL. (**b**) AuNPs of 40 nm had an actual measured LOD of 0.5 pg/mL and a dynamic range of 0.5–128 pg/mL. (**c**) AuNPs of 60 nm had an actual measured LOD of 0.125 pg/mL and a dynamic range of 0.125–32 pg/mL. The original data has been placed in Appendix A.

**Figure 9 pharmaceutics-15-01493-f009:**
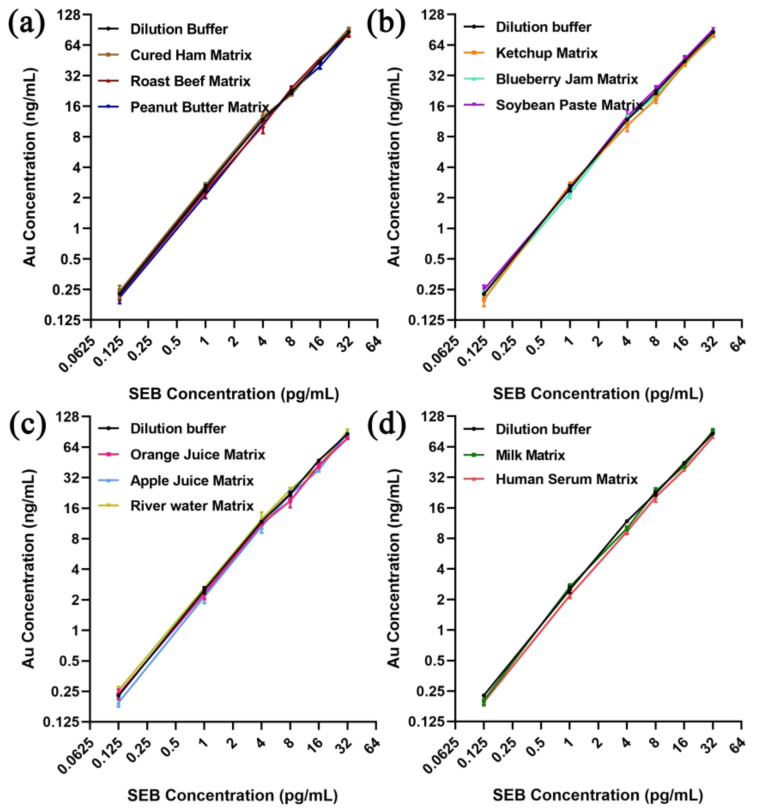
Recoveries of SEB detection by ALISA in various matrices. (**a**) Recoveries of SEB detection by ALISA in solid food matrices including cured ham, roast beef, and peanut butter. (**b**) Recoveries of SEB detection by ALISA in semi-solid food matrices including ketchup, blueberry jam, and soybean paste. (**c**) Recoveries of SEB detection by ALISA in liquid matrices including orange juice, apple juice, and river water. (**d**) Recoveries of SEB detection by ALISA in milk and human serum. The original data has been placed in Appendix A.

**Table 1 pharmaceutics-15-01493-t001:** Summary of maximum absorption peak before and after AuNP modification.

Particle Size (nm)	Plasmon Absorption Peak (nm)		
	AuNPs	AuNPs/HS-PEG-COOH	AuNPs/HS-PEG-COOH/ FMU-SEB-1
15	519.5	521.5	525.5
40	523.5	526	534.5
60	533	536	541

**Table 2 pharmaceutics-15-01493-t002:** The comparison of newly established methods for SEB detection.

Methods	Linear Range	Detection Limit	Detection Time (min)	Detection Sample	Recovery Rates (%)	Ref.
Lateral flow assay	N/A	6 pg mL^−1^	more than 60	Milk, canned meat, baby food, canned mushrooms	N/A	[35]
Chemiluminescence immunoassay	3.12–50 ng mL^−1^	1.44 ng mL^−1^	more than 60	Milk, water	82.5–95.2	[8]
FRET	0.001–1 ng mL^−1^	0.3 pg mL^−1^	more than 50	Milk	86.0–110.0	[36]
SERS	2–100 pg mL^−1^	1.3 pg mL^−1^	more than 120	Milk	88.2–91.67	[37]
Bio-barcode & real-time PCR	0.001–100 ng mL^−1^	0.269 pg mL^−1^	more than 150	Milk, metamorphosed milk, water, orange juice	89.17–110.29	[38]
Colorimetric assay	0.5–50 μg mL^−1^	0.5 ng mL^−1^	50	Milk, cheese, ice-cream, chicken, pastries	N/A	[39]
ALISA	0.125–32 pg mL^−1^	0.125 pg mL^−1^	about 150	River water, milk, orange juice, apple juice, human serum, ketchup, blueberry, soybean, roast beef, cured ham, peanut butter	92.7–95.0	ALISA method

**Table 3 pharmaceutics-15-01493-t003:** Intra-assay and inter-assay of SEB ALISA.

Concn Concentration of SEB (pg/mL)	Mean Measured Concentration (pg/mL)	CV (%)
intra-assay (*n* = 8)		
2	1.84 ± 0.08	4.4
8	7.70 ± 0.41	5.3
20	18.82 ± 1.36	7.2
inter-assay (*n* = 8)		
2	1.84 ± 0.23	12.3
8	8. 11 ± 0.88	10.8
20	20.07 ± 2.20	11.0

**Table 4 pharmaceutics-15-01493-t004:** Recovery in dilution buffer of SEB ALISA.

Spike Levels (pg/mL)	Mean Measured Concentration (pg/mL)	Mean Recovery (%)
2	1.90 ± 0.10	95.0 ± 4.8
8	7.50 ± 0.48	93.8 ± 6.0
20	18.55 ± 1.48	92.7 ± 7.4

## Data Availability

All obtained data are presented in this article.

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
