# Peer review of "Development of a Gold Nanoparticle-Linked Immunosorbent Assay of Staphylococcal Enterotoxin B Detection with Extremely High Sensitivity by Determination of Gold Atom Content Using Graphite Furnace Atomic Absorption Spectrometry"

_pharmaceutics, 2023, doi:10.3390/pharmaceutics15051493_

Round 1
Reviewer 1 Report
The manuscript presents an original development that significantly enhances sensitivity of microplate immunoassay. It accords to basic demands of MDPI, but needs some revisions:
1. The use of graphite furnace atomic absorption spectrometry is an essential feature of the proposed development. It really leads to significant lowering the detection limit, but can be realized only with the help of special high-cost equipment. This fact should be clear for readers to avoid misunderstanding. So I propose to include the indication of the graphite furnace atomic absorption spectrometry into the title, whereas the specification «in various matrices» can be excluded due to low information output. The named above limitations of the GFAAS use should be specified in Abstract and in the Conclusions.
2. Time of the assay is its important parameter that should be indicated in the Abstract.
3. Please comment the following assays of SEB that are stated as extremely sensitive ones: DOI 10.1016/j.scitotenv.2022.159977 – LOD = 3.43 fg/mL; DOI 10.1016/j.foodchem.2022.133271 – 0.103 × 10(−6) ng/mL.
4. To evaluate novelty of the proposed approach, earlier works with the combination of immunoassays and atomic absorption spectrometry for labels detection should be considered in the Introduction independently on target antigen (i.e. not only for SEB).
5. Why the authors used two-step synthesis for AuNPs with a diameter of 40 nm and 60 nm? Different simple one-step techniques such as citrate (Frenz) synthesis can be applied for this purpose.
6. Please comment the chosen proportion between FMU-SEB-1 and modified AuNPs in the course of their conjugation (Section 2.4).
7. Lines 149-153. Please describe manipulations with aqua regia more detail. At what stages plastic was used, and at what stages was glass used? How was the completeness of liquid taking from microplate wells ensured?
8. Please use «g» values for centrifugation regimes instead of «rpm» values (lines 165, 171), as well as the «rpm» conditions may be reproduced only with the use of the same rotor.
9. Figs. 4, 5 do not present UV parts of spectra and so cannot be named as data of UV-visible spectroscopy.
10. Please specify normalization (excluding of scattering component) for spectra at Figs. 4, 5.
11. For grounded conclusion about LOD, «three sigma» criterion should be applied.
12. The consideration of a working range for quantitative measurements will be more correct that the actually presented linear range.(Linear range can potentially start from zero point.)
Reviewer 2 Report
The presented work is devoted to development of sandwich immunoassay of staphylococcal enterotoxin B using novel detection approach. The work is well planned and clearly written, but the topic of the publication has nothing to do with the scope of PHARMACEUTICS, nor with the topic of a special issue dedicated to the prevention, diagnosis and treatment of cancer. So, I recommend to transfer the manuscript to BIOSENSORS or FOODS
Besides, a number of issues should to be addressed before publication.
L 47 The authors probably meant the lowest limit of detection, not the highest. Please change it.
L 48-50 Specify the much higher values of LOD for cited methods
Reference 20. Substitute authors’ names by authors’ surnames
L77 Correct “Methods”
L128 Specify dialysis membrane and volume of water
L 151-152 Please, detail the procedure. Where were solutions of aqua regia from a 96-well plate placed and where and why were they boiled before burning, etc?
Fig 1. Indicate on the scheme where are FMU-141 SEB-1 and FMU-141 SEB-2 antibodies.
Mention domain specificity of Abs or provide references if available.
L168 Please, specify if you mean a water sample when you use the word "river".
L210. Change black line to red line.
Table 1 Revise the title
L 300, 310 Combine two Tables 2 into one Table 2.
Check for typos
L 77 Methods
L 216 blue
L 290 inraassay
Reviewer 3 Report
The authors have conducted research in a highly relevant field of detecting Staphylococcal enterotoxin B (SEB) in various matrices with extremely high sensitivity.
However, there are a few areas that require improvement before publishing the article in the journal.
1. It would be beneficial to have an in-depth comparison of the achieved analytical performance with the best methods published in recent years, particularly those that have the best limits of detection. The authors should include a table of comparison with other methods (the analysis time, dynamic range, and other parameters should be included).
2. The abstract needs to be reworked to present a single, logical presentation, instead of being divided into subheadings.
3. I would recommend focusing on the best of the most modern practices in the introduction. I would recommend not limiting to the phrase that the detection methods have "the highest limit of detection (LOD) of SEB reported was 0.01 ng/mL [12]", where [12] is a 2010 work.
In particular, more sensitive methods have since emerged, such as those:
Xu Y. et al. (2019). Food Chem., 283, 338-344.
Bragina V.A. et al. (2019) Anal. Chem., 91(15), 9852-9857.
Shen H. et al (2022). ACS App. Mat. & Interfaces, 14(3), 4637-4646.
4. Interestingly, at zero concentration, the calibration curves start from almost zero signal along the OY axis. Is this due to the complete absence of non-specific binding, or is it because the authors mathematically subtracted "base" values?
5. Figure 8 is built from the results of measuring only one sample for each concentration (n = 1). This is discouraging as it is impossible to calculate the detection limit according to the criteria accepted for such calculations (according to the 3.3 sigma criterion). The calibration curves should be constructed using multiple replicate samples (n > 2) for each concentration, so that standard deviations can be calculated, and then the analytical limit of detection can be calculated using a three-sigma criterion.
6. It is recommended that all values and all initial data used for graphs and tables be presented as supplementary material (including the results of each individual experiment for those cases where n > 1).
Reviewer 4 Report
1. Aqua regia is toxic if inhaled, and causes severe burns and eye damage. The authors used aqua regia to dissolve AuNPs and even boiled the dissolved solution for 30 min before measurement of gold atom. Due to the use of aqua regia, is there anything the authors do to protect themselves during experiments? Even this ALISA has high sensitivity, but it will be difficult to be practically applied?
2. TEM is unclear, therefore, FTIR before and after adding antibody will be better to confirm the conjugation of antibody onto AuNPs.
3. Please check the resolution of figures, they are not in good condition. All figures should be in high resolution.
4. Experiments should be repeated 3 times and curve with error bar should be shown in fig 8.
5. Conclusions section should be rewritten to emphasize the advantage and disadvantage of this ALISA.
English language is fine
Round 2
Reviewer 1 Report
The manuscript has been successfully revised and now may be published
Author Response
Dear Professor,
Thanks for your approval of our work, and we are appreciated your kindly efforts in reviewing the article.
Best wishes,
Li Fan
Reviewer 3 Report
I would like to express my gratitude to the authors for their hard work in improving the manuscript. I am pleased to see that the authors have taken into account most of my suggestions, which has greatly enhanced the quality of the paper.
However, I would like to kindly suggest that new table S1 be included in the main text of the manuscript, accompanied by an appropriate description. This would allow for greater clarity and accessibility for readers.
Author Response
Dear Professor,
Thanks for your kindly efforts in reviewing the article. According to your suggestion, we have placed the table S1 in the main text of the manuscript and the refences are updated. The description of the table is as follows:
Table 2 lists the newly established methods for SEB detection, including their linear ranges, detection limits, detection times, etc. Noticeably, compared to other methods, ALISA is highly sensitive.
All the revisions to the manuscript are marked up using the “Track Changes” function.
Thanks again for your kindly consideration.
Best wishes,
Li Fan
Reviewer 4 Report
I would recommend this manuscript for publication.
Author Response
Dear Professor,
Thank you for taking your valuable time to review our paper. We appreciate your constructive suggestions and comments on the articles. Thank you for your efforts on the quality of our paper.
Best wishes,
Li Fan